# ROBOWHEEL: A HELICAL DATA ENGINE FROM REAL-WORLD HUMAN DEMONSTRATIONS FOR CROSS-DOMAIN ROBOTIC LEARNING

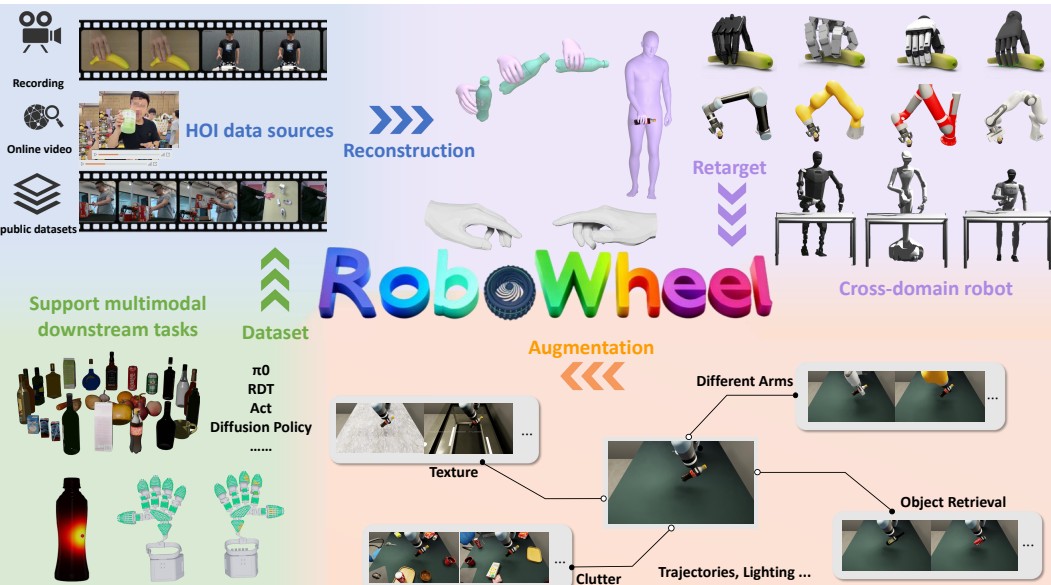

Figure 1: The *RoboWheel* helical data engine. Our pipeline could process hand-object interaction (HOI) videos from diverse sources (*e.g.*, online recordings and public datasets) through high-fidelity reconstruction to recover physically consistent trajectories. The reconstructed motions are retargeted to cross-domain robotic embodiments (*e.g.*, arms, dexterous hands, humanoids), and enhanced with multi-modal data augmentations (*e.g.*, object retrieval, texture, trajectories). This generates a large-scale dataset with multimodal observations (RGB-D, poses, contacts), supporting training for various vision-language-action (VLA) and imitation learning models (*e.g.*, ACT, Diffusion Policy).

## ABSTRACT

We introduce *RoboWheel*, a helical data engine that converts in-the-wild human hand–object interaction (HOI) videos into training-ready supervision for cross-morphology robotic learning. From monocular RGB/RGB-D inputs, we perform high-precision HOI reconstruction and enforce physical plausibility via a reinforcement learning optimizer that refines hand–object relative poses under contact and penetration constraints. The reconstructed, contact-rich trajectories are then retargetted to cross-domain embodiments, robot arms with simple end-effectors, dexterous hands, and humanoids, yielding executable actions and rollouts. To scale coverage, we build a simulation-augmented framework on Isaac Sim with diverse domain randomization (body variants, trajectories, object replacement, background changes, hand motion mirroring), which expands observations and labels while preserving contact semantics. This process forms an end-to-end pipeline from video → reconstruction → retargeting → augmentation → data acquisition, closing the loop for iterative policy improvement. Across vision-language-action and imitation-learning settings, *RoboWheel*-generated data provides reliable supervision and consistently improves task performance over baselines, enabling direct use of Internet HOI videos (hand-only or upper-body) as labels for scenario-specific training. We further assemble a large-scale multimodal dataset combining multi-camera captures, monocular videos, and public HOI corpora, and demonstrate transfer on dexterous-hand and humanoid platforms.

## 1 INTRODUCTION

Embodied agents learn most effectively when supervision reflects how humans actually interact with the physical world. However, obtaining contact-rich, robot-usable supervision at scale remains notoriously difficult. Existing pipelines typically rely on prelabeled, curated human video datasets or studio motion capture, which limits coverage, diversity, and transfer across different embodiments and tasks. Notably, the Internet contains an immense reservoir of hand–object interaction (HOI) videos (hand-only or upper-body) that include rich manipulation strategies, but these signals are rarely converted into training-ready data for robots due to reconstruction noise, physical implausibility, and embodiment mismatching.

We revisit this problem through the lens of modern perception. Human/hand and object motion estimators (*e.g.* SMPL-H/MANO parameters for articulated hands/bodies and 6D object pose/mesh trackers) now extract stable geometry and motion from monocular RGB/RGB-D inputs. However, the raw outputs remain inadequate for control because contact estimates may be inconsistent, interpenetrations can occur under occlusion, and trajectories often fail to respect robot kinematics. These challenges reveal a persistent gap between "*what can be reconstructed*?" and "*what a robot can effectively execute or learn from*?".

Nevertheless, turning Internet-scale HOI videos into reliable robotic supervision is far from trivial. The pipeline must overcome challenges at both the data source level and during reconstruction. On the one hand, existing paradigms for collecting robot training data—teleoperation or simulated demonstrations—are costly and biased or fail to capture real-world physical and perceptual distributions. On the other hand, extracting high-precision hand–object interactions from monocular video introduces issues of camera/world-frame inconsistency, severe occlusions, unreliable object pose estimation, and violations of physical plausibility. We detail these challenges in Appendix A.

To migrate this gap, we introduce *RoboWheel*, a helical data engine from real-world human demonstrations for cross-domain robotic learning. *RoboWheel* turns in-the-wild HOI videos into training-ready supervision for various robotic embodiments. Using state-of-the-art hand, whole-body, and object motion estimation methods from monocular RGB/RGB-D video, the pipeline consolidates the motion into a unified, robust framework for hand–object joint optimization and cross-embodiment retargeting, outputting control trajectories in both operational (end-effector) and joint spaces. Technically, our system mainly includes the following four parts, (i) Reconstruct hand/body and object motions from video; (ii) Multi-stage, physically grounded optimizer—projection losses for 2D consistency, SDF-based collision/contact penalties, and an RL-guided refinement that maximizes plausibility of the hand–object relative pose under stability/reachability priors; (iii) we retarget the refined trajectories to multiple morphologies (robot arms with simple grippers, dexterous hands, humanoids) via kinematic/dynamic constraints to produce executable actions; and (iv) we run simulation-based data augmentation in Isaac Sim with domain randomization (left-right-hand mirroring, embodiment variants, object replacement, background changes), preserving contact semantics while expanding observations.

Before delving into details, we list our key contributions as follows.

- **Precision, physically plausible HOI reconstruction and cross-domain retargeting.** A contact-consistent HOI reconstruction framework from monocular RGB/RGB-D, combining SOTA hand/whole-body/object motion estimation with multi-stage physical optimization. It integrates cross-embodiment retargeting, providing scalable supervision across diverse robot embodiments (arms, dexterous hands, humanoids) with executable trajectories in operational and joint spaces.
- **Simulation-augmented data flywheel.** A dynamic augmentation and domain randomization pipeline based on Isaac Sim ( embodiment variants, object replacement, background variation, hands mirroring, etc.) conditioned on HOI. This data flywheel is validated on mainstream VLA and imitation-learning settings to enhance robustness and scalability in robotic learning.
- **Large-scale multimodal dataset.** Thousands of high-precision (augmented to 150k+) sequences from an in-house multi-view mocap pipeline, public HOI datasets, and curated online videos, including robot actions, hand–object motions, tactile signals, multi-view observations, and task descriptions, provide a rich, scalable resource for robotic learning and HOI models.

## 2 RELATED WORK

**HOI datasets and monocular reconstruction.** High-precision HOI annotations in the wild remain costly, as most public 3D HOI datasets rely on multiview rigs or motion capture (MoCap)

systems for accurate hand-object geometry (Chao et al., 2021; Hampali et al., 2020; 2021; Taheri et al., 2020; Wang et al., 2024). Large egocentric video corpora like Grauman et al. (2024) use head-mounted cameras to avoid MoCap but lack frame-accurate 3D HOI geometry for reconstruction Grauman et al. (2022). Recent whole-body motion datasets such as Zhang et al. (2025) scale to millions of SMPL-X frames but are not dedicated HOI datasets and offer limited hand-object contact supervision. On the algorithmic side, Chen et al. (2025c) reconstructs objects by fusing pixel-aligned features with 3D hand geometry in a transformer-based coarse-to-fine point cloud decoder, yielding dense object geometry with high frame fidelity, while Fan et al. (2024) jointly reconstructs articulated hands and objects using compositional SDF and contact constraints. These methods, however, are generally limited to single-frame or in-contact scenarios and struggle with approach/withdrawal phases, generalizability, occlusion, low video resolution, and varying hand movement speeds. Recently, more generalizable approaches (Prakash et al., 2023; Yang et al., 2023; Qu et al., 2023) have used data-driven priors; for instance, Yang et al. (2023) introduces diffusion-guided, per-video optimization to enhance robustness under occlusion, albeit at the cost of heavier computation and the need for short clips.

**Embodied models and scalable data for generalist manipulation.** Generalist *vision–language–action* policies pretrained on large video and robot corpora to enable instruction following and out-of-distribution generalization across tasks and embodiments (Brohan et al., 2022; Zitkovich et al., 2023; Kim et al., 2024; Black et al., 2025). In parallel, imitation- and diffusion-based visuomotor learning emphasize stable training and multimodal action distributions, from classic action-diffusion policies to large diffusion foundation models that scale to bimanual control Chi et al. (2023); Liu et al. (2024). To reduce data and hardware barriers, low-cost bimanual teleoperation systems provide dense demonstrations for fine-grained skills (Zhao et al., 2023), while object-/pose-centric representations and semantic flows improve cross-object generalization and pose awareness Chen et al. (2025b). At the dataset/benchmark layer, dual-arm generators and domain-randomized platforms supply scalable supervision with unified evaluation (Mu et al., 2025; Chen et al., 2025a); open-instruction rearrangement benchmarks probe 6-DoF reasoning under language guidance (Ding et al., 2024); and video-driven pipelines synthesize long-horizon tasks directly from Internet videos (Ye et al., 2025). Recent work on task-centric *latent actions* further mitigates embodiment mismatch by learning instruction-conditioned action spaces transferable across robots (Bu et al., 2025).

Due to page limitations, we leave *Robotic Learning from Human Demonstration* discussion in Appendix B.

## 3 METHOD

### 3.1 SYSTEM OVERVIEW

We build a systematic pipeline covering in-the-wild hand-object interaction(HOI) videos into robot-usable supervision data. An overview of our pipeline is illustrated in Fig. 2.

### 3.2 HAND MOTION AND OBJECT RECONSTRUCTION FROM RGB(D) VIDEOS (STAGE I)

**Problem setup.** Given video frames $\{I_t\}_{t=1}^{T}$, our goal is to recover metrically consistent trajectories and parametric representations of both the articulated hand and the manipulated object in the same world coordinate. Concretely, the state of the hand pose at time $t$ is,

$$\mathbf{h}_t = (\theta_h(t), \mathbf{R}_h^w(t), \mathbf{t}_h^w(t)), \tag{1}$$

where $\theta_h(t)$ is the hand pose, $\mathbf{R}_h^w(t)$ and $\mathbf{t}_h^w(t)$ are the global transform and wrist of hands in the world coordinate. The object state is the rigid 6D pose tied to its (scale-resolved) geometry, $\mathbf{p}_t = T_o^w(t) \in \mathtt{SE}(3)$, defines the location and rotation of the object.

**Human and hand motion recovery.** Our method initially determines whether a clip implies *hand-only* or *whole-body* motion. For the *hand-only* case, we estimate $\mathbf{h}_t$ per frame using Pavlakos et al. (2024). Otherwise, we estimate the SMPL-H parameters via Zhang et al. (2025) and directly produce the world coordinate body pose $\theta_b(t)$ and the shape $\beta_b$, equivalently extracting the hand state $\mathbf{h}_t$.

**Object reconstruction and pose estimation.** We ground the manipulated object, obtaining the per-frame mask $m_t$ and depth $D_t$ (predicted by Piccinelli et al. (2025) or RGB-D) in the video. Conditioned on semantic cues, we use a multiview 3D generator $\mathcal{G}$ (Zhao et al., 2025) to produce an

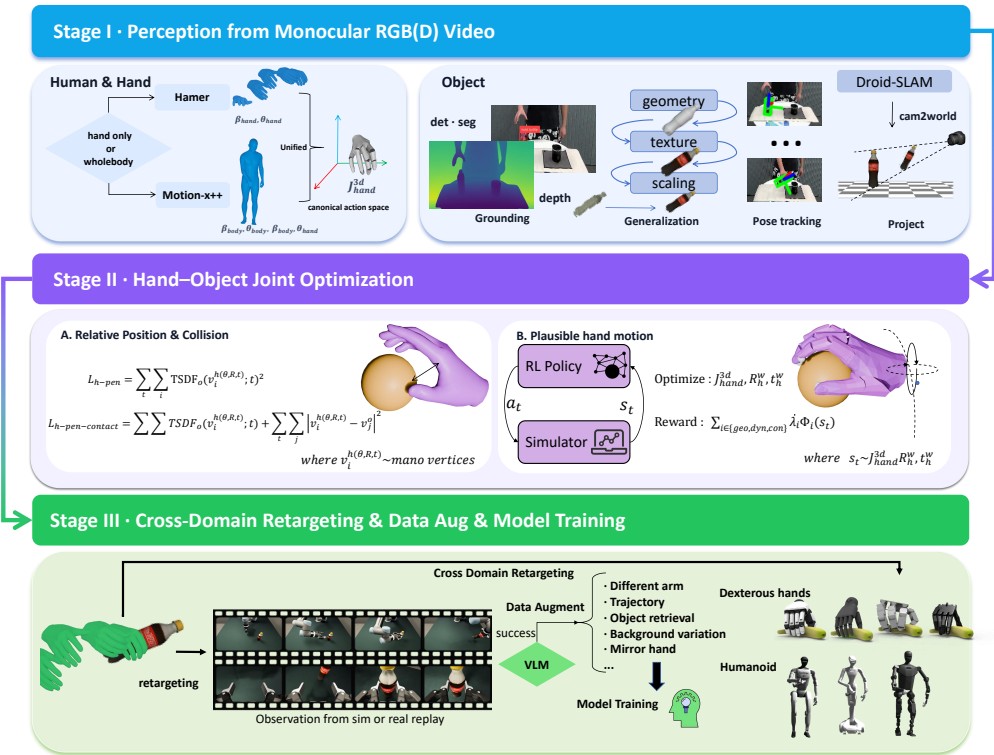

Figure 2: For monocular RGB(-D) input, we first estimate hand, whole-body, and object motion. A multi-stage optimizer then applies: (I) projection losses aligning to 2D evidence; (II) collision/contact constraints with temporal regularization; and (III) RL-guided refinement improving physical plausibility and reachability. Refined trajectories are retargeted to diverse embodiments (grippers, dexterous hands, humanoids) and exported in operational/joint space. Finally, HOI-conditioned domain randomization in Isaac Sim expands observations, closing a helical data loop.

unscaled textured mesh $\hat{M}_o$. Then we recover the metric scale of the manipulated object by back-projecting the depth map inside the mask to a point set $\mathcal{P}_t = \{X_c(p) = D_t(p)\,K^{-1}\tilde{p} \mid p \in m_t\}$, aggregating as $\mathcal{P} = \bigcup_t \mathcal{P}_t$. Letting $\mathrm{diag}(\cdot)$ denote the diagonal of the axis-aligned bounding box $\mathrm{AABB}(\cdot)$, we set $M_o$ as the estimated rescaled object,

$$M_o \;=\; s_o\,\hat{M}_o, \qquad s_o \;=\; \|\mathrm{diag}(\mathrm{AABB}(\mathcal{P}))\|_2 / \|\mathrm{diag}\!\left(\mathrm{AABB}(\hat{M}_o)\right)\|_2. \tag{2}$$

where $s_o$ is the estimated scale factor. With $(M_o, M_t, D_t)$, a correspondence-driven tracker $\mathcal{F}(\cdot)$ (Wen et al., 2024) estimates the pose stream of the camera frame object $T_o^c(t)$.

**Project to a unified action space.** To eliminate viewpoint-dependent inconsistencies in real-world HOI videos, we first estimate the camera intrinsics $K$ and the camera-to-world transformation $T_c^w = (R_{wc}, t_{wc})$ using Teed & Deng (2021). This allows us to transform all reconstructed hand-object interactions to the world coordinate system. We then align the resulting trajectories to a canonical action space $\mathcal{A}$ by constructing a reference frame based on body joint positions, ensuring consistency across heterogeneous sources. For detailed transformation steps, please refer to Appendix D.

### 3.3 JOINT OPTIMIZATION FOR HAND AND OBJECT INTERACTION (STAGE II)

**Problem setup.** Given monocular frames $\{I_t\}_{t=1}^T$ with estimated intrinsics $K$ and extrinsics $(\mathbf{R}_{wc}, \mathbf{t}_{wc})$, we seek temporally consistent hand motion and object trajectories in the same world coordinate. As defined in Sec. 3.2, the hand state is $\mathbf{h}_t = \left(\theta_h(t), R_h^w(t), t_h^w(t)\right)$, and the object state $\mathbf{p}_t = T_o(t)$ have been coarsely initialized by the last stage. We then jointly optimize $\{\mathbf{h}_t, \mathbf{p}_t\}_{t=1}^T$ to (i) prevent hand-object interpenetration and (ii) enforce physically plausible and temporally stable contact.

**Phase (A): Physics- and contact-consistent refinement.** Let $\phi_o(\mathbf{x}; t)$ be a watertight object SDF (positive outside), and $V_h$ the point cloud of hand vertices. The optimization pipeline is as follows. First, we optimize hand parameter $\mathbf{t}_h$ to avoid penetration between object and hand-palm.

$$\mathcal{L}_{\text{h-pen}} = \sum_t \sum_i \left[ \max\left(0, -\phi_o(v_h^i; t)\right) \right]^2, \quad i \in \left\{ i \mid v_h^i \in V_h^{\text{palm}}(\mathbf{t}_h) \right\}.$$

Then, we optimize the hand parameter $\mathbf{R}_h$, $\mathbf{t}_h$ and $\boldsymbol{\theta}_h$ to avoid penetration between object and hand with better grasping pose.

$$\mathcal{L}_{\text{h-pen-contact}} = \eta_{\text{pen}} \sum_t \sum_i \left[ \max\left(0, -\phi_o(v_h^i; t)\right) \right]^2 + \eta_{\text{contact}} \sum_t \sum_j \left\| v_h^j - v_o^k \right\|_2^2$$
$$+ \eta_{\text{smooth}} \sum_t \left( \left\| \Delta^2 \mathbf{t}_h(t) \right\| + \left\| \log\left(\mathbf{R}_h(t-1)^\top \mathbf{R}_h(t)\right) \right\|_F^2 \right),$$

where $i \in \left\{ i \mid v_h^i \in V_h(\boldsymbol{\theta}_h, \mathbf{R}_h, \mathbf{t}_h) \right\}$, $j \in \left\{ j \mid v_h^j \in K\text{-closest vertices to object} \right\}$, $v_o^k$ is the closest object vertice to $v_h^j$.

**Phase (B): Residual RL refinement for reachability and plausibility.** Inspired by the residual control framework (Li et al., 2025a), a reinforcement learning (RL) refinement process is conducted in simulation to achieve physically plausible hand-object poses and ensure reachability on robots. Given the human-object interaction (HOI) state $s_t = \left(h_t, p_t, \dot{h}_t, \dot{p}_t, \mathcal{C}_t\right)$ within the physical environment, a residual learning strategy is applied to refine the trajectories of both the hand and the object. The reward function $r_t$ is defined as:

$$r_t = \underbrace{\lambda_{\text{geo}} \Phi_{\text{geo}}\left(-\|\Delta h_t\| - \|\Delta p_t\|\right)}_{\text{geometric reward}} + \underbrace{\lambda_{\text{dyn}} \Phi_{\text{dyn}}\left(-\|\Delta \dot{h}_t\| - \|\Delta \dot{p}_t\|\right)}_{\text{kinematic reward}} + \underbrace{\lambda_{\text{con}} \Phi_{\text{con}}\left(\mathcal{C}_t\right)}_{\text{contact reward}},$$

where $\Phi$ denotes the reward function and $\Delta$ the error between simulated and target states.

### 3.4 CROSS-DOMAIN RETARGETING (STAGE III)

Based on the physically plausible joint HOI reconstruction in Section 3.2, we obtained physically plausible trajectories $\{h_t, p_t\}_{t=1}^T$ and ensured stable hand-object contacts. We aim to retarget these to heterogeneous robot embodiments—industrial arms, dexterous hands, and humanoids.

**Robot arms.** Given accurate 3D hand joints, we retarget hand poses into executable end-effector poses $\{T_g(t), g(t)\}_{t=1}^T$ for a parallel-jaw gripper (Fig. 3). Inspired by Kjellstrom et al. (2008b), we implement two complementary orientation constructions depending on whether the *whole hand* (palm-involved) or *only finger tips* dominate the contact geometry. *Whole-hand* retargeting builds a stable palm frame from MCP joints to suppress fingertip jitter; *finger-only* mapping aligns to a hand-intrinsic frame and uses the index–thumb chord to define the gripper axis.For the detailed algorithm, please refer to the Appendix F.To assess the state of the gripper, we employ CoTracker Karaev et al. (2024) to track the motion trajectories of key points on the manipulated object. The gripper state is determined based on the displacement of these key points.A key ad-

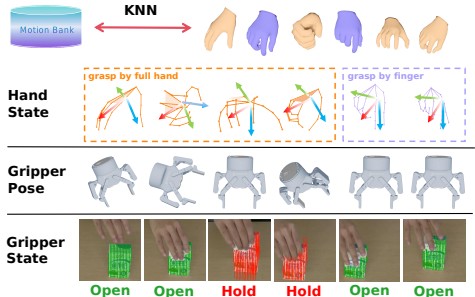

Figure 3: We categorize hand poses into two types: grasp by full hand and grasp by finger, each associated with a specific retargeting method. Using the KNN algorithm, we classify hand poses and perform orientation mapping. For gripper state, we track target object keypoints. If displacement occurs, the gripper is considered closed; otherwise, it is open.

vantage of this keypoint-oriented approach lies in its robustness to the significant visual ambiguity caused by severe occlusions of the object mask during manipulation.

Beyond retargeting to simple gripper-based arms, our high-fidelity HOI reconstruction enables transferring to more complex embodiments such as dexterous hands and humanoid robots,as shown in figure4. For dexterous hands, we retarget the reconstructed hand motions to the joint space of target robotic hands using kinematic similarity and contact-preserving constraints. This allows us to generate fine-grained finger motion trajectories that maintain functional grasp semantics.

**Dexterous hands and humanoids.** For whole-body human demonstrations, we extend retargeting to humanoid platforms by leveraging full-body SMPL-H estimates. The resulting motion sequences are adapted to humanoid joint trees through inverse kinematics and dynamics-aware optimization, ensuring physical plausibility and intent preservation.

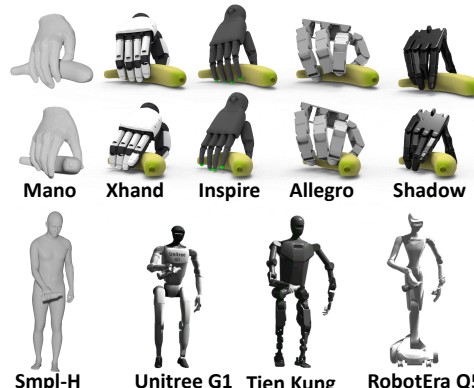

**Mano   Xhand   Inspire   Allegro   Shadow**

**Smpl-H   Unitree G1   Tien Kung   RobotEra Q5**

Figure 4: Cross-domain embodiment retargeting.

With this unified retargeting framework, we expand the scalability and diversity of *RoboWheel* data engine. Each human demonstration is automatically transduced into multiple, semantically aligned training episodes spanning a broad range of robot embodiments—from parallel-jaw grippers to dexterous multi-fingered hands and humanoids. This mechanism materially amplifies the effective yield of every collected video by multiplying cross-embodiment supervision. By constructing a large-scale, cross-domain corpus in this manner, *RoboWheel* furnishes directly usable supervision for training generalist robotic policies that transfer skills and knowledge across heterogeneous hardware.

3.5   DATA AUGMENTATION IN SIMULATION (STAGE III)

We enhance observation diversity in simulation through HOI-conditioned domain randomization while preserving the contact semantics essential for control. All HOI-to-workspace transformations are defined in the canonical action space $\mathcal{A}$, thereby ensuring consistent contact frames and approach directions across randomized environments.

**Different types of arm retargeting.** Given an executable end-effector (EE) trajectory $\{T_g(t)\}$ produced by our retargeting method, we generate observations for heterogeneous arms, as illustrated in Fig. 5 . We instantiate in Isaac Sim five widely used 6–7 DoF robotic arms as simulation assets: *UR5/UR5e*, *Franka Emika Panda*, *KUKA LBR iiwa 7*, *Kinova Gen3*, and *Rethink Robotics Sawyer*. HOI-derived 6D EE trajectories $T_g(t) \in \text{SE}(3)$, $t = 1, \ldots, T$, are mapped into feasible joint trajectories using cuRobo's GPU-accelerated inverse kinematic (IK) backend (Sundaralingam et al., 2023). For each robotic arm, at every timestep we invoke IK solver with the target pose $T_g(t)$. The solver returns a feasible joint configuration:

$$q_t = \arg\min_q \ \mathcal{C}_{\text{goal}}(T_g(t), q) \quad \text{s.t.} \ q_{\min} \preceq q \preceq q_{\max}, \ \mathcal{C}_{\text{coll}}(q) \leq 0, \tag{3}$$

where $\mathcal{C}_{\text{goal}}$ is cuRobo's pose reaching cost and $\mathcal{C}_{\text{coll}}$ is the self-collision constraint. To encourage temporal consistency, we use the previous solution $q_{t-1}$ as the IK seed when invoking the solver.

Episodes that pass the replay check retain the original HOI intent (*e.g.*, grasp/carry/place/pour) while providing embodiment diversity in joint space. We export both the joint-space commands $\{q_t\}_{t=1}^T$ (arm and gripper included) and aligned operational-space labels per robot, enabling multi-morphology policy training from the same HOI source.

**Object retrieval and replacement.** We build a large object library by combining Zhao et al. (2025) generations with in-house scans; each asset includes a watertight mesh, texture, category tag, and a canonical pose. For a source episode with object mesh $M_o$ and object pose stream $\{T_o(t)\}$, we retrieve top-$K$ substitutes $\tilde{M} = \{\tilde{M}_k\}$ using a fused similarity,

$$\mathcal{S}(M_o, \tilde{M}) = \alpha\, \text{CD}(\hat{M}_o, \hat{\tilde{M}}) + \beta\left(1 - \text{IoU}_{\text{AABB}}\right)$$
$$+ \gamma \langle \phi_{\text{sem}}(M_o), \phi_{\text{sem}}(\tilde{M}) \rangle,$$

where $\hat{\ }$ denotes unit AABB normalization, CD is the symmetric Chamfer distance on surface samples, $\text{IoU}_{\text{AABB}}$ measures coarse shape compatibility, and $\phi_{\text{sem}}$ are text–shape embeddings.

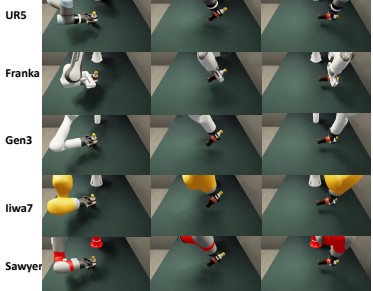

Figure 5: *RoboWheel* retargets hand motion to different kinds of robot arms.

To ensure replay compatibility on a retrieved substitute, we align principal axes and bind the same maximum AABB and canonical pose definition as the source $M_o$. Under this binding, the original EE plan and the hand–object interaction geometry remain consistent, so the control trajectory can be directly replayed on geometrically/semantically matched novel objects (*e.g.*, mug $\leftrightarrow$ cup-with-handle, box $\leftrightarrow$ carton) while preserving task intent.

Table 1: Modalities and scale comparison. "✗" and "✓" denote presence and absence respectively.

| Dataset | Tactile/ Force | Contact State | Robot Arm | Real-World Human Operation | # Trajectories | Object Info Granularity |
|---|---|---|---|---|---|---|
| **Robowheel-150K (ours)** | ✓ | ✓ | ✓ | ✓ | *150328* | RGB(D) + 6-DoF object pose + Assets |
| GRAB Taheri et al. (2020) | ✗ | ✓ | ✗ | ✓ | 1334 | Body & 6-DoF object pose & Contact maps |
| HO3D (v3) Hampali et al. (2021) | ✗ | ✗ | ✗ | ✓ | 68 | 6-DoF object pose + YCB assets |
| DexYCB Chao et al. (2021) | ✗ | ✗ | ✗ | ✓ | 1,000 | RGB-D + 6-DoF object pose + YCB assets |
| HO-Cap Wang et al. (2024) | ✗ | ✗ | ✗ | ✓ | ~64 | Hand/object 3D shape + pose (multi-view) |
| DROID Khazatsky et al. (2024) | ✗ | ✗ | ✓ | ✓ (tele-op) | 76,000 | RGB(+Depth), lang, robot states |
| LIBERO Liu et al. (2023) | ✗ | ✗ | ✓ | ✗ (simulation) | 366 | Sim assets & states; benchmark tasks (130) |
| UCSD Kitchen Yan et al. (2023) | ✗ | ✗ | ✓ | ✗ (robot runs) | 150 | RGB + joint states/torques; no object assets |
| TACO Li et al. (2023) | ✗ | ✗ | ✗ | ✓ | 2,500 | Precise hand–object **meshes** + action labels |

**Trajectory augmentation.** Informed by Xue et al. (2025) and tailored to our setting, we represent each demonstration as a trajectory $\tau = \{(T_g(t), g(t))\}_{t=1}^{T}$, where $T_g(t) = (R(t), p(t)) \in \mathrm{SE}(3)$ denotes the EE pose with orientation $R(t)$ and translation $p(t)$, and $g(t)$ is the gripper command. The trajectory is partitioned into object-centric segments $\{\tau^{(k)}\}$, each labeled by a contact state $c^{(k)} \in \{\texttt{hold}, \texttt{open}\}$. Instead of re-planning trajectories, we augment them as follows.

*(i)* For interaction segments ($c^{(k)} = \texttt{hold}$), we apply an object-frame rigid transform $T_o \in \mathrm{SE}(3)$ to each waypoint:

$$\tilde{T}_g(t) = T_o T_g(t), \qquad \tilde{g}(t) = g(t).$$

Let $R_\Delta := \mathrm{Rot}(T_o)$. To maintain continuity without motion-plan regeneration, the same EE orientation change is applied to non-interaction segments (see (ii)), and the orientation change induced by $R_\Delta$ is kept small for IK feasibility and repeatable execution.

*(ii)* For each non-interaction segment ($c^{(k)} = \texttt{open}$), we linearly remap the translational path and set the EE orientation as $\tilde{R}(t) = R_\Delta R(t)$. Let $p_s, p_e$ be the original endpoints and $\hat{p}_s, \hat{p}_e$ the remapped anchors: the anchor adjacent to a transformed interaction segment is fixed by that segment, while the opposite anchor is chosen within a predefined reachable set. With $\alpha_t \in [0, 1]$ denoting the normalized progress along the original segment from $p_s$ to $p_e$,

$$\tilde{p}_t = \hat{p}_s + \alpha_t(\hat{p}_e - \hat{p}_s) + \Big[ p_t - \big( p_s + \alpha_t(p_e - p_s) \big) \Big].$$

## 4 DATASET

Based on the *RoboWheel* system, we assembled a large-scale multimodal HOI-to-robot dataset built by converting heterogeneous human videos into robot-usable episodes and augmenting them across embodiments. The dataset includes (i) in-the-wild Internet HOI videos, (ii) public HOI datasets, and (iii) our self-collected mocap high-precision capture system. We convert these data using *RoboWheel* HOI reconstruction pipeline—omitted when HOI annotations already exist and perform cross-domain retargeting to generate observation streams suitable for replay in both simulation and real-robot settings. Combined with cross-domain augmentation, this enables model training across diverse robot embodiments. Comparing with previous datasets, in Tab. 1, shows the following strengths and unique properties.

**Diverse HOI reconstruction modalities.** Robowheel dataset contains approximately 150k frames drawn from internet clips, reprocessed public HOI corpora, and our studio captures. Each episode includes synchronized multi-view RGB/RGB-D observations, per-frame MANO parameters in the world frame (pose/shape with global orientation and translation), 6-DoF object pose with a textured asset (mesh/texture ID), contact states and partially

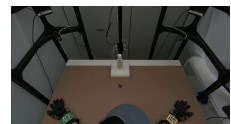
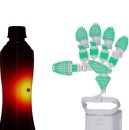
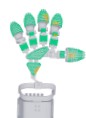

Data Collection Setup    Tactile Signal for hand and object

Figure 6: Data collection setup and tactile information presentation

available tactile signals(shown in Fig. 6), and fine-grained language task descriptions. For more details, please refer to Appendix I.

**Diverse embodiment modalities.** From the reconstructed HOI trajectories, we provide retargeted robot control labels for multiple embodiments—arms, dexterous hands, and humanoids—including operational-space ($\mathrm{SE}(3)$ end-effector) trajectories and joint-space commands. These labels are temporally aligned with the observations and are suitable for embodied models training.

Table 2: HOI reconstruction quality comparison. **Object surface:** CD (cm) = bidirectional Chamfer distance; F5/F10 (%) = F-score at 5/10 mm. **Hand:** Hand jitter (cm/s$^2$) = time-avg. norm of frame-to-frame wrist/palm acceleration (30 FPS, 2nd-order diff.); W-MPJPE (mm) = wrist-relative MPJPE after aligning the wrist. **Rel. pose consistency:** std of $T_{\mathrm{rel}}(t) = T_h^{-1}(t)T_o(t)$ in translation (cm) / rotation (deg).

| Method | Object | | | Hand | | Rel. pose consistency ↓ | |
|---|---|---|---|---|---|---|---|
| | CD (cm)↓ | F5 (%)↑ | F10 (%)↑ | Hand jitter (cm/s$^2$)↓ | WA-MPJPE (mm)↓ | Trans (cm)↓ | Rot (deg)↓ |
| HORT | 8.9 | 55.0 | 83.0 | 3.35 | 19.92 | 3.54 | - |
| DiffHOI | 7.2 | 59.6 | 78.1 | 4.59 | 20.21 | 4.51 | - |
| HOLD | 7.5 | 53.2 | 77.9 | 3.47 | 20.59 | 2.44 | - |
| **Ours** | 5.1 | 63.4 | 89.1 | 0.92 | 7.81 | 0.26 | 1.9 |

Table 3: Real-world task performance grouped by difficulty (easy *vs.* hard).

| Real-world Tasks | Diff. | ACT
tele. \| RW | DP
tele. \| RW | RDT
tele. \| RW | Pi0
tele. \| RW | RDT+5kRW
RW | Pi0+5kRW
RW |
|---|---|---|---|---|---|---|---|
| Pick up milk | | 15.0% \| 0.0% | 20.0% \| 15.0% | 55.0% \| 30.0% | 70.0% \| 65.0% | 70.0% | 80.0% |
| Lift wooden cup | Easy | 0.0% \| 0.0% | 45.0% \| 25.0% | 75.0% \| 45.0% | 65.0% \| 55.0% | 70.0% | 70.0% |
| Place milk | | 35.0% \| 0.0% | 50.0% \| 35.0% | 70.0% \| 50.0% | 80.0% \| 55.0% | 85.0% | 80.0% |
| Restore bowl | | 0.0% \| 0.0% | 5.0% \| 0.0% | 65.0% \| 65.0% | 60.0% \| 60.0% | 75.0% | 75.0% |
| Average | | 12.5% \| 0.0% | 30.0% \| 18.8% | 66.3% \| 47.5% | 68.8% \| 58.8% | 75.0% | 76.3% |
| Move banana | | 0.0% \| 0.0% | 5.0% \| 0.0% | 55.0% \| 20.0% | 40.0% \| 15.0% | 65.0% | 60.0% |
| Upright milk | Hard | 0.0% \| 0.0% | 0.0% \| 15.0% | 45.0% \| 30.0% | 60.0% \| 50.0% | 60.0% | 75.0% |
| Pour cola | | 0.0% \| 0.0% | 0.0% \| 10.0% | 35.0% \| 35.0% | 25.0% \| 35.0% | 40.0% | 55.0% |
| Tip teacup | | 0.0% \| 0.0% | 0.0% \| 0.0% | 5.0% \| 15.0% | 35.0% \| 25.0% | 25.0% | 45.0% |
| Average | | 0.0% \| 0.0% | 1.3% \| 6.3% | 35.0% \| 25.0% | 40.0% \| 31.3% | 47.5% | 58.8% |

# 5 EXPERIMENT

## 5.1 HOI RECONSTRUCTION QUALITY

We evaluate HOI reconstruction quality on Wang et al. (2024) with common metrics in Tab. 2 and Fig. 7. All methods receive the same camera parameters and object meshes for a fair comparison. We compare *RoboWheel* with HORT (2025c), HOLD (2023), and DiffHOI (2023).

## 5.2 *RoboWheel* PERFORMANCE ON EMBODIED TRAINING DATA

Here we show how *RoboWheel*-derived embodied data support downstream tasks and how well learned skills transfer with data augmentation.

**Performance on different VLA/IL models.** To study how Robowheel reconstructions translate to downstream control, we benchmark ten household tasks grouped by difficulty (Easy/Hard) and evaluate several VLA/IL algorithms (ACT, DP, RDT, Pi0). We eval-

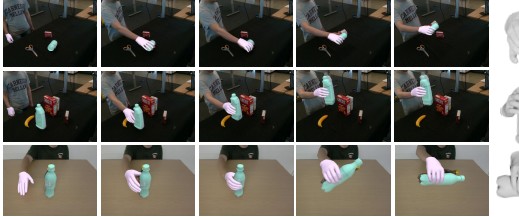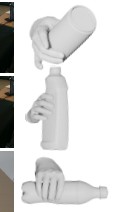

Figure 7: HOI reconstruction results of *RoboWheel* .Whether the data comes from public HOI datasets (*e.g.*, DexYCB) or not, *RoboWheel* can achieve high-precision HOI reconstruction.

uate each algorithm under three distinct training regimes: (i) *Teleoperation Demonstrations* (Tele.), (ii) an equivalent number of trajectories generated by *RoboWheel* (RW), and (iii) a two-stage regime involving pre-training on an additional 5k RoboWheel trajectories followed by task-specific fine-tuning (denoted as RDT+5kRW and Pi0+5kRW). Per-task success rate(%) is reported in the same real setup in Tab. 3. Macro averages within each difficulty group are reported.

As shown in Tab. 3 and Fig. 8, VLA policies (Pi0 & RDT) pretrained with 5k *RoboWheel* data achieved the highest success rates, demonstrating a remarkable performance improvement. The impact of this improvement is more evident in tasks of higher complexity. Notably, when training or fine-tuning these VLA/IL methods with an identical number of training episodes from teleoperation or *RoboWheel*, policies trained solely on *RoboWheel* data achieve performance comparable to those trained on teleoperation data, despite the sim-to-real gap. This result highlights the effectiveness of *RoboWheel* data for augmentation.

The underlying reason for this is twofold. First, *RoboWheel* provides precise HOI reconstruction, enabling the generation of trajectories with accuracy approaching that of teleoperation, ensuring

Figure 8: Real-world Performance on 4 Tasks

Table 4: Task performance in unseen scenarios with RDT.

| Real-world Tasks | Unseen Object | | Clutter Objects | | Unseen Background | |
|---|---|---|---|---|---|---|
| | RW | RW-aug | RW | RW-aug | RW | RW-aug |
| Lift wooden cup | 4/10 | 5/10 | 4/10 | 4/10 | 0/10 | 4/10 |
| Place milk | 5/10 | 6/10 | 5/10 | 6/10 | 1/10 | 3/10 |
| Upright milk | 3/10 | 3/10 | 4/10 | 4/10 | 3/10 | 5/10 |
| Pour cola | 3/10 | 3/10 | 3/10 | 4/10 | 2/10 | 4/10 |
| Average | 3.75/10 | 4.25/10(+0.5) | 4.00/10 | 4.50/10(+0.5) | 1.50/10 | 4.00/10(+2.5) |

Table 5: Robot **direct replay** success rate $\uparrow$ (%).

| Task | Ours | GAT-Grasp | yoto |
|---|---|---|---|
| Lift wooden cup | 87.5 | 25.0 | 75.0 |
| Place bowl | 100 | 62.5 | 100 |
| Upritght milk | 100.0 | 50.0 | 62.5 |
| Pour cola | 100 | 62.5 | 75.0 |
| move banana | 87.5 | 62.5 | 87.5 |
| Tip teacup | 87.5 | 37.5 | 00.0 |
| Macro avg | 91.7 | 50.0 | 66.7 |

effective transfer from simulation to real-world environments. Second, data augmentation is applied to *RoboWheel*, and its broader data distribution helps the policy mitigate the negative impact of the visual domain gap, endowing the policies with enhanced robustness.

**Generalization on unseen object, unseen background, and clutter.** We probe generalization of the four household skills under three shifts: *unseen object instances*, *unseen backgrounds*, and *cluttered scenes*. We compare models trained on Robowheel only (RW) versus RW with augmentation (RW-aug: object category & clutter & background). Each cell shows *successes/trials* for two independent runs, as shown in Tab. 4. This setup isolates whether reconstruction-driven data and simple augmentations suffice to transfer the skill to new objects, backgrounds, and cluttered environments.

When trained on the *RoboWheel* data without augmentation, the finetuned RDT still manages to achieve some successful trials when encountering unseen objects or inferring in cluttered environments. However, when there are significant changes in the observations (e.g., changing the background), the model experiences catastrophic performance degradation. In contrast, the fine-tuned RDT with augmented RoboWheel data shows a significant improvement in handling new observations, particularly in the unseen background setting, where the success rate increased by 25%.

### 5.3 REAL-ROBOT REPLAY PERFORMANCE COMPARE

Under the same hand-motion input, we retarget the hand motions to a two-finger gripper using different existing methods and execute the same set of tasks, using success rate to quantify the robustness of each mapping scheme. As evidenced by the comparisons in Tab. 5 and Fig. 19, our retargeting approach consistently attains higher success rates across tasks, and its definition of gripper orientation is more robust—supporting flexible execution of tasks under diverse hand-gesture regimes.

## 6 CONCLUSION

Our helical data engine *RoboWheel* is designed to transform real-world hand-object interaction videos into cross-domain robotic learning supervision. By leveraging high-fidelity reconstruction, multi-stage optimization, and cross-embodiment retargeting, *RoboWheel* offers a scalable framework for generating training data that is physically consistent and directly applicable to diverse robotic platforms. We demonstrated its effectiveness through an innovative data pipeline that includes simulation-augmented data augmentation and domain randomization, allowing for the generation of a large-scale multimodal dataset that supports various VLAs and imitation learning models.

## ETHICS STATEMENT

This research adheres to ethical guidelines in all aspects of the study. *RoboWheel* is a system that converts in-the-wild human hand-object interaction (HOI) videos into embodied supervision for cross-domain robotic learning. In our research, we used videos from the internet, but their usage is strictly limited to academic purposes. Any harmful use is not intended or encouraged.

## REPRODUCIBILITY STATEMENT

To reporduce our retargeting results in both real-world and simulated environments, please consult the Appendix F, where we present the pseudocode for the retargeting process. Regarding the real-world performance evaluation of the four VLA/IL policies, the code is fully sourced from the open-source repository, with implementation specifics provided in the Appendix J.1.

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

## A  PROBLEM AND CHANNLLENGE

### A.1  LIMITATIONS OF DATA SOURCES FOR ROBOT LEARNING

Building robust, general-purpose robotic systems requires large-scale, diverse, high-quality training data that remain aligned with real-world distributions and respect physical plausibility. Existing data acquisition paradigms broadly fall into two categories—teleoperation and generative/simulation-based approaches. The former is costly and difficult to scale, subject to operator bias and inconsistency, and struggles to cover long-horizon, contact-rich scenarios; the latter, while easy to expand, exhibits pronounced sim-to-real gaps, insufficient physical modeling, and perceptual domain shift, leading to distributional mismatch with the real world and limited transferability and verifiability.

### A.2  CHALLENGES IN HIGH-PRECISION HAND AND OBJECT RECONSTRUCTION

Despite substantial progress in Human–Object Interaction (HOI), extracting high-quality HOI data from monocular video remains challenging, particularly for real-world applications that demand both high precision and scalability. Existing HOI reconstruction pipelines are constrained in several respects: 1) many methods reconstruct in the camera coordinate frame; when camera intrinsics/extrinsics are unknown, transforming to the world frame that underlies robot action spaces often yields pronounced, physically implausible trajectory jitter; 2) object pose estimation typically hinges on strong prerequisites—e.g., access to object models/assets and metric depth—and even when such conditions are satisfied, occlusions by the manipulating hand or other objects frequently induce large pose-tracking errors; and 3) the reconstructed relative hand–object configurations often violate physical plausibility—indeed, non-negligible errors are observed even in HOI datasets captured with motion-capture systems and multi-view camera rigs.

## B  RELATED WORK

**Robotic Learning from Human Demonstration**  Early work in vision-based programming by demonstration mapped human hand poses to robot grasps directly from images, establishing a pipeline from grasp recognition to example-based robot execution Kjellstrom et al. (2008a). More recent systems leverage richer HOI signals: Zhou et al. (2025) extract binocular hand-motion cues from human videos, compress trajectories into keyframes with coordination masks, augment demonstrations geometrically, and train a bimanual diffusion policy that executes long-horizon dual-arm tasks and generalizes across scenes. Complementarily, Wang et al. (2025) treat human gestures as structured priors, retrieving grasp affordances from HOI memories and transferring them to novel objects, yielding robust performance in single-object and cluttered settings. At scale, cross-embodiment corpora and models (Open X-Embodiment/RT-X) demonstrate positive transfer across heterogeneous robots, motivating retargetable supervision from human interactions (Vuong et al., 2023). Specialized transfer frameworks extend this idea to dexterous bimanual manipulation via residual learning (Li et al., 2025b), while whole-body humanoid control benefits from motion-tracking pipelines distilled into guided diffusion policies that enable versatile downstream behaviors (Truong et al., 2025). Together, these threads indicate a viable route from human HOI video to robot-usable policies via reconstruction, retargeting, and data augmentation.

## C  NOTATION USED THROUGHOUT THE PAPER.

| Symbol | Meaning | Notes / Space |
|---|---|---|
| $\{I_t\}_{t=1}^T$ | Input frame sequence | RGB / RGB-D |
| $K$ | Camera intrinsics | Known or estimated |
| $T_c^w$ | Camera pose in world frame | $SE(3)$; often $(R_{wc}, t_{wc})$ |
| $\Pi(\cdot)$ | Pinhole projection operator | 3D $\rightarrow$ pixel coordinates |
| $\Pi_{xz}(\cdot)$ | Projection onto $xz$-plane | Removes $y$ component |
| $\mathbf{h}_t$ | Hand state at time $t$ | $(\theta_h(t), R_h^w(t), t_h^w(t))$ |
| $\theta_h(t)$ | Hand kinematic parameters (MANO/SMPL-H) | Joint angles / shape |
| $R_h^w(t), t_h^w(t)$ | Wrist pose in world (rot/trans) | $SO(3)$ and $\mathbb{R}^3$ |
| $\mathbf{p}_t$ | Object state at time $t$ | $T_o^w(t) \in SE(3)$ |
| $T_o^w(t) = (R_o(t), t_o(t))$ | Object pose in world (rot/trans) | $SE(3)$ |
| $M_o, \hat{M}_o$ | Object mesh (metric / up-to-scale) | Triangle mesh + texture |
| $s_o$ | Object scale factor | From depth–mesh alignment |
| $m_t, D_t$ | Object mask and depth map at $t$ | Segmentation and depth |
| $P_t, P$ | Back-projected points (per-frame / global) | From $D_t$ and $K$ |
| $\mathrm{AABB}(\cdot)$ | Axis-aligned bounding box | Use diagonal via $\mathrm{diag}(\cdot)$ |
| $A$ | Canonical action space | Unified facing/origin |
| $T_w^A = (R^{w \rightarrow A}, t^{w \rightarrow A})$ | World-to-action-space transform | Coordinate re-alignment |
| $\Delta^2(\cdot)$ | Second-order temporal difference | Jitter suppression |
| $\rho(\cdot)$ | Robust loss (e.g., Geman–McClure) | For reprojection errors |
| $\mathrm{IoU}(\cdot, \cdot)$ | Intersection-over-Union | Contour/overlap consistency |
| $d(\cdot, \cdot)$ | Point–set / point–surface distance | For proximity/attraction |
| $V_h, V_o$ | Hand mesh vertices / object surface points | Geometry sets |
| $\phi_h(\mathbf{x}; t)$ | Hand signed distance field (SDF) | Positive outside (as used here) |
| $TSDF_o$ | object truncated signed distance function | negative inside and zero else |
| $\tilde{\mathbf{q}}_i(t)$ | World coords of sampled object surface point | $R_o(t) \mathbf{q}_i^{\mathrm{loc}} + t_o(t)$ |
| $\mathcal{N}_t$ | Near-contact candidate set | $|\phi_h| < \tau_{\mathrm{band}}$ or Top-$K$ |
| $\| \cdot \|_F$ | Frobenius norm | For rotation log etc. |
| $\log_{SO(3)}(\cdot)$ | Lie-group log map on $SO(3)$ | Rotation discrepancy |
| $d_{SE(3)}(\cdot, \cdot)$ | Geodesic distance on $SE(3)$ | Pose discrepancy |
| $\mathrm{E}_\pi[\cdot]$ | Expectation under policy $\pi$ | RL objective |
| $r_t$ | Instantaneous reward | Weighted components |
| $\psi_{\mathrm{limits}}(\cdot)$ | Joint-limit margin reward | Prefer mid-range |
| $\pi_\theta$ | Residual policy network | Added to $a^{\mathrm{IK}}$ |
| $a_t^{\mathrm{IK}}$ | Inverse-kinematics baseline action | ManipTrans-style residual control |
| $T_{\mathrm{rel}}(t)$ | Relative pose $T_h^{-1}(t)\, T_o(t)$ | Hand–object relative pose |
| ADD-S | Average Distance (symmetric objects) | Pose error metric (cm) |
| $R_g, p_g$ | Gripper pose (rot/trans) | From hand keypoints |
| $q_t^{(r)}$ | Joint configuration of robot $r$ at $t$ | From bounded-rate IK |
| $\phi_{\mathrm{lim}}(q)$ | Joint-limit penalty | IK constraint term |
| $m(q)$ | Manipulability | Avoid singularities |
| $S = \mathrm{diag}(-1, 1, 1)$ | Left–right mirror matrix | About sagittal plane |
| $R_y(\pi)$ | Rotation about $y$ by $\pi$ | Used with $S$ to set facing |
| $\varphi_{\mathrm{sem}}(\cdot)$ | Semantic embedding (text–shape) | For semantic similarity |
| $\gamma^t$ | Discount factor power | RL return discounting |

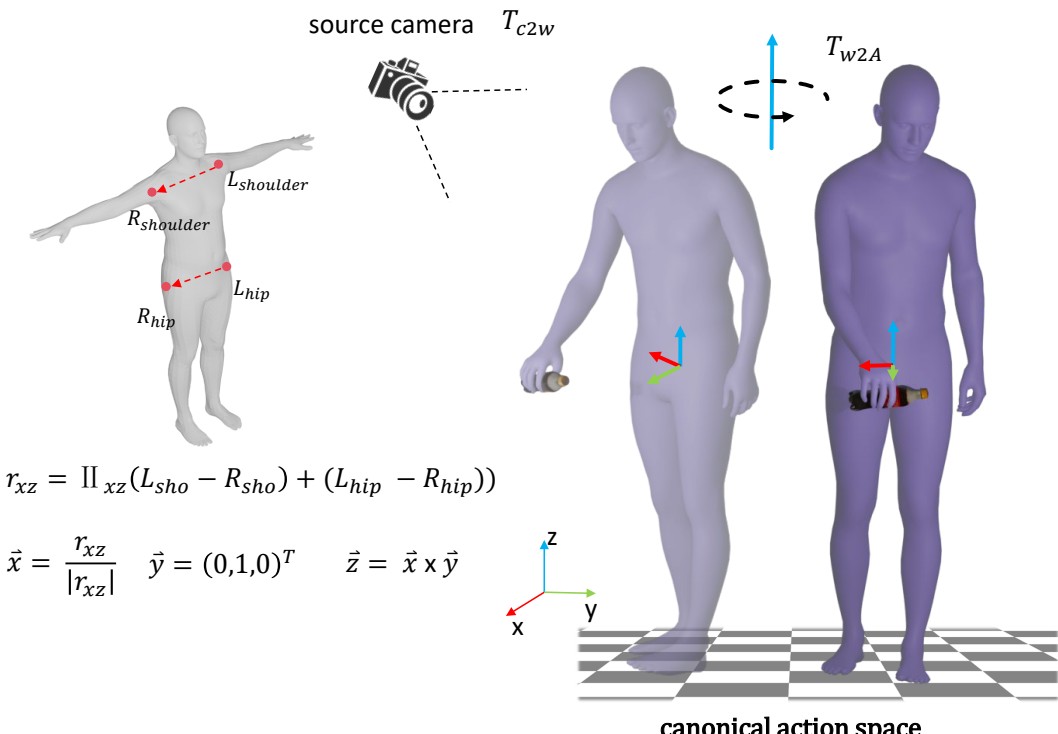

$$r_{xz} = \Pi_{xz}(L_{sho} - R_{sho}) + (L_{hip} - R_{hip}))$$

$$\vec{x} = \frac{r_{xz}}{|r_{xz}|} \quad \vec{y} = (0,1,0)^T \quad \vec{z} = \vec{x} \times \vec{y}$$

**canonical action space**

Figure 9: Our pipeline decouples the reconstructed hand-object motions from the specific camera viewpoint of the source video. We first lift the reconstructions to a consistent world frame using the camera-to-world transformation $T_{c2w}$. Subsequently, we normalize these trajectories into a canonical action space via $T_{w2A}$. This two-step alignment ensures that the retargeted actions maintain a consistent approach direction and kinematic interpretation across all robotic embodiments.

## D  CONVERT HOI MOTION TO CANONICAL ACTION SPACE

Real-world HOI videos are captured under arbitrary viewpoints, which leads to view-dependent reconstructions of both hand and object. To eliminate this inconsistency, we adopt a lifting procedure, as shown in Fig. 9. In the first step, all reconstructed HOI results are transformed from their respective camera coordinate systems into the same world coordinate system. In the second step, the trajectories in the world coordinate system are further normalized into a unified canonical action space, ensuring that interaction trajectories from heterogeneous sources become retargetable.

**Step 1: Estimate $(K, T_c^w)$ and lift to the world frame.** We assume a static-camera prior and estimate camera intrinsics $K$ and a (time-invariant) camera-to-world transform $T_c^w = (R_{wc}, t_{wc})$ with DROID-SLAM Teed & Deng (2021), optimizing sparse/semi-dense reprojection together with temporal smoothness:

$$\min_{K, T_c^w} \sum_{t,i} \left\| \Pi\left(K, (T_c^w)^{-1}; X_i\right) - u_{i,t} \right\|_2^2 + \lambda \left( \|\Delta t_{wc}\|_2^2 + \left\| \log_{SO(3)}(R_{wc}^\top R_{wc}^+) \right\|_2^2 \right), \quad (4)$$

where $(\cdot)^+$ denotes the next keyframe. We adopt the keyframe solution as the clipwise $T_c^w$. We also evaluated DPVO and COLMAP and found DROID-SLAM more stable on our setting. With the fixed $T_c^w$ for each clip, we could project our estimated $\mathbf{h}_t, \mathbf{p}_t$ convert to the world frame:

**Step 2: Align to canonical action space $\mathcal{A}$.** Given the left/right hip and shoulder positions $p_{\text{hip}}^{L/R}, p_{\text{sho}}^{L/R}$ in the world coordinate system $\{\mathcal{Z}\}$, we first compute a lateral reference vector on the $xz$-plane:

$$v_{\text{lat}} = \Pi_{xz}\left((p_{\text{hip}}^L - p_{\text{hip}}^R) + (p_{\text{sho}}^L - p_{\text{sho}}^R)\right).$$

We then construct the canonical frame $\mathcal{A}$ with the following conventions:

- $z_{\mathcal{A}}$ is aligned with the scene up direction (gravity / ground normal).

- $y_{\mathcal{A}}$ is determined by the dominant interaction direction (e.g., the average hand$\rightarrow$object approach vector).
- $x_{\mathcal{A}}$ is obtained by the right-hand rule, ensuring orthonormality.

After orthonormalization, these axes form the rotation matrix $R_{w\rightarrow\mathcal{A}} \in \mathrm{SO}(3)$. Finally, we center the action trajectory at the object position (by default, at the first salient frame $t_0$), giving the translation $t_{w\rightarrow\mathcal{A}} = -R_{w\rightarrow\mathcal{A}}\, t_o^w(t_0)$ and the resulting rigid transformation $T_w^{\mathcal{A}} = (R_{w\rightarrow\mathcal{A}}, t_{w\rightarrow\mathcal{A}})$.

## E  DATA AUGMENTATION

Here, we present various data augmentation strategies and additional details applied to more tasks.

### E.1  RATARGETING TO DIFFERENT ROBOT ARM

Here, we present additional results from the data augmentation module, demonstrating motion retargeting for HOI reconstruction to different robotic arms. Figure ?? and Figure 11 show the retargeted motions for the tasks "flip milk," "pour water," and "place milk" to the *UR5/UR5e*, *Franka Emika Panda*, *KUKA LBR iiwa 7*, *Kinova Gen3*, and *Rethink Robotics Sawyer* arms, respectively.

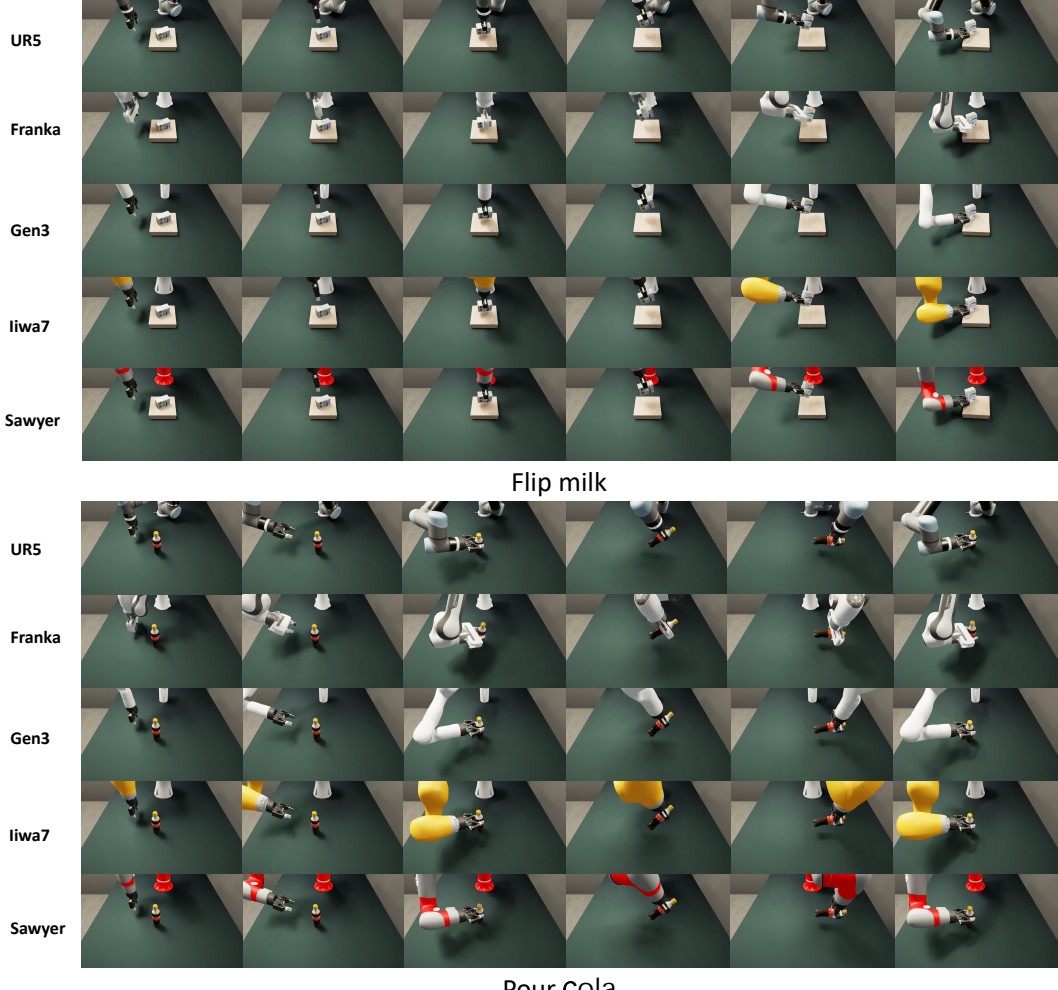

Figure 10: Visualization of robot arm augmentation:flip milk and pour Water

UR5

Franka

Gen3

Iiwa7

Sawyer

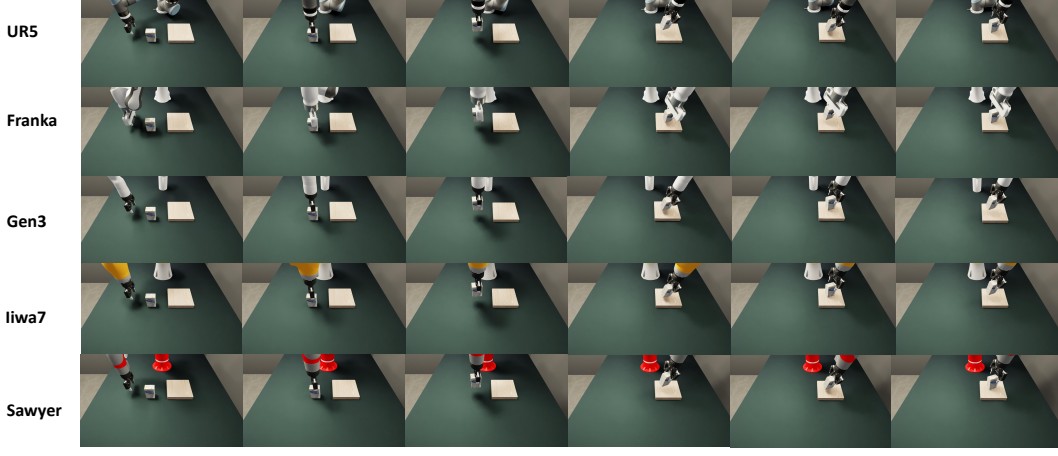

Place milk

Figure 11: Visualization of robot arm augmentation:place milk

## E.2 BACKGROUND VARIATION(TEXTURE)

As shown in Fig. 12We apply scene-level visual randomization to diversify the pixel distribution while keeping task dynamics and contact semantics unchanged: *i*) workspace and background appearance randomization (e.g., tabletop, backsplash) via texture and normal-map swaps, and adjustment of basic PBR parameters (albedo/roughness); *ii*) illumination randomized using parametric light sources (cylinder and sphere lights) with variations in spatial placement, intensity, color, color temperature, and emission radius, enabling a broad range of plausible lighting conditions; *iii*) clutter regime ranging from empty scenes to heavy distractors, with randomly sampled object positions and orientations placed collision-free outside the robot's swept volume via rejection sampling; *iv*) mild camera intrinsics/extrinsics jitter consistent with prior calibration to emulate plausible view changes.

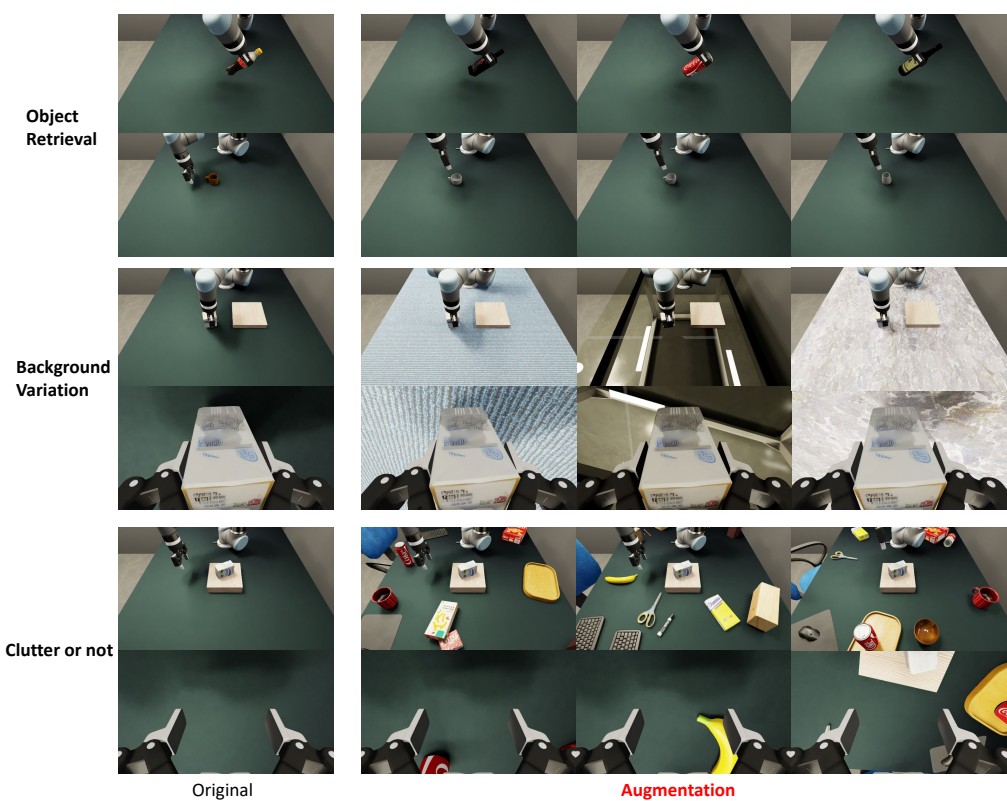

Figure 12: Diverse background texture augmentation in *RoboWheel*.

### E.3 OBJECT RETRIEVAL

Our object-retrieval augmentation strategy successfully enables the transfer of manipulation skills to novel objects in simulation. By replacing the original object with a retrieved counterpart that shares high geometric and semantic similarity, and initializing it in the same canonical pose, the robot can reliably execute the same action trajectory. Visually confirmed in Figure 13, Figire 14, and Figure 15 for tasks including "pour water", "tip tea cup", and "place box".

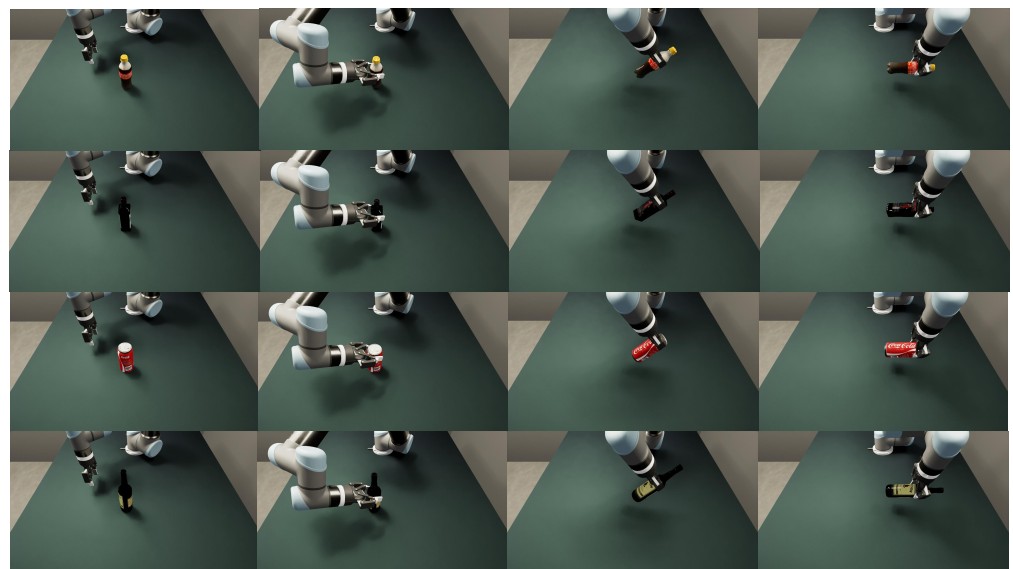

Figure 13: Object Retrieval augmentation

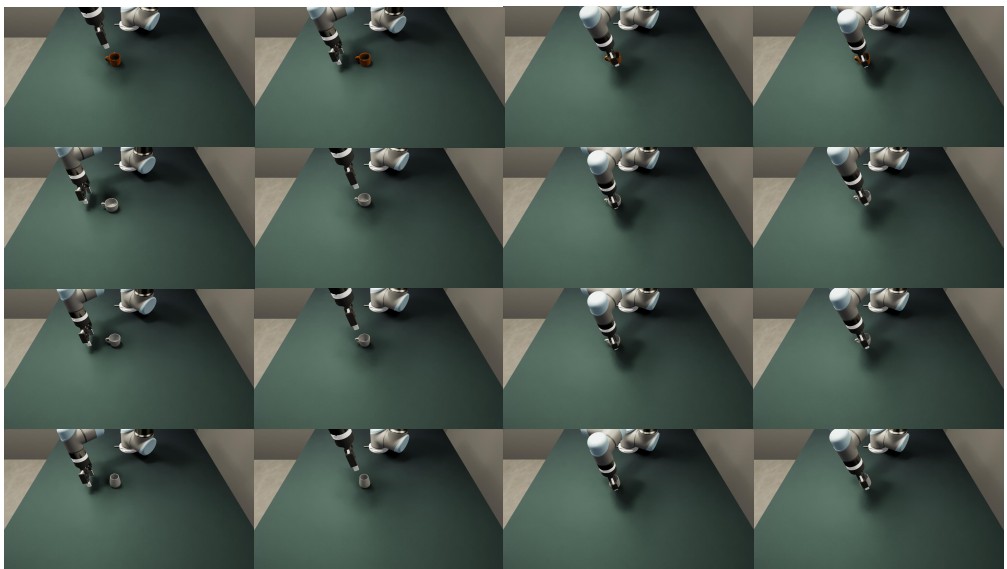

Figure 14: Object Retrieval augmentation

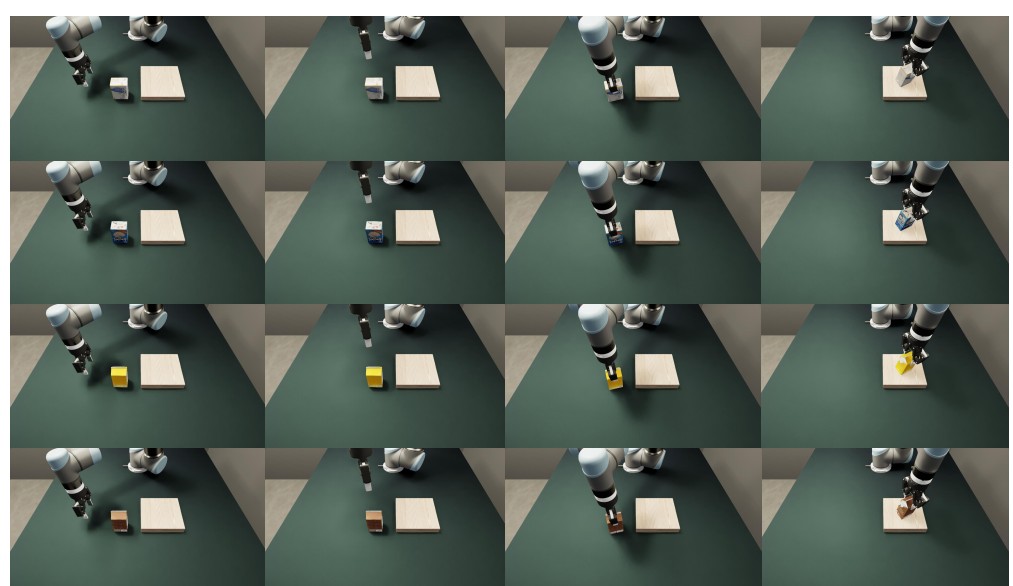

Figure 15: Object Retrieval augmentation

### E.4  HAND MIRROR

**Motivation.** Many daily manipulations are left–right symmetric up to a sagittal-plane reflection; mirroring increases trajectory diversity without changing task semantics.

**Operator.** Let the sagittal reflection be $S = \mathrm{diag}(-1, 1, 1)$. For positions, $p'(t) = S\,p(t)$. A pure reflection is improper for orientations, so we compose a $\pi$-rotation about the $y$-axis to recover a proper rotation:

$$R'(t) \;=\; S\,R(t)\,S \cdot R_y(\pi), \qquad \det\!\big(R'(t)\big) = +1. \tag{5}$$

We mirror *both* hand and object about the same plane so that $T'_{\mathrm{rel}}(t) = T'_h(t)^{-1}T'_o(t) = T_{\mathrm{rel}}(t)$, preserving contact frames and approach vectors. Gripper chirality and finger-axis signs are flipped consistently.

**Safeguards.** We exclude actions whose handedness encodes semantics (e.g., threaded fasteners), detected via a non-zero screw component about the task $z$-axis exceeding $\tau_{\mathrm{screw}}$. Mirrored rollouts must pass the replay check on a reference arm before inclusion.

**Algorithm 1** Whole-Hand (Palm-Involved) Gripper Pose Construction

**Input:** wrist $\mathbf{k}_{\text{wri}}$, index MCP $\mathbf{k}_{\text{ind}}^{\text{mcp}}$, ring MCP $\mathbf{k}_{\text{ring}}^{\text{mcp}}$

**Output:** Gripper pose $(R_g, \mathbf{p}_g)$.

$\mathbf{w} \leftarrow \mathbf{k}_{\text{wri}}, \mathbf{i} \leftarrow \mathbf{k}_{\text{ind}}^{\text{mcp}}, \mathbf{r} \leftarrow \mathbf{k}_{\text{ring}}^{\text{mcp}}$
  *// Extract keypoints*
$\mathbf{o} \leftarrow (\mathbf{w} + \mathbf{i} + \mathbf{r})/3$   *// palm origin*
$v_x \leftarrow (\mathbf{r} - \mathbf{w})$   *// X-axis direction*
$\bar{\mathbf{x}} \leftarrow v_x/(\text{NORMALIZE}(v_x) + 10^{-8})$
  *// Normalized X-axis*
$v_z \leftarrow \text{CROSS\_PRODUCT}(\mathbf{i} - \mathbf{w}, \ \mathbf{r} - \mathbf{w})$
  *// Z-axis (palm normal)*
$\bar{\mathbf{z}} \leftarrow v_z/(\text{NORMALIZE}(v_z) + 10^{-8})$
  *// Normalized Z-axis*
$\bar{\mathbf{y}} \leftarrow \text{CROSS\_PRODUCT}(\bar{\mathbf{z}}, \bar{\mathbf{x}})$
  *// Y-axis direction*
$\bar{\mathbf{z}} \leftarrow \text{SIGN}(\diamond) \cdot \bar{\mathbf{z}}$
$R_g \leftarrow \text{CONCATENATE}([\bar{\mathbf{x}}, \bar{\mathbf{y}}, \bar{\mathbf{z}}])$
  *// Rotation matrix*
$\mathbf{p}_g \leftarrow \mathbf{o} + d_z \bar{\mathbf{z}}$   *// Position*
**Return** $(R_g, \mathbf{p}_g)$

**Algorithm 2** Finger-Only (Pinch/Precision) Gripper Pose Construction

**Input:** index TIP $\mathbf{k}_{\text{ind}}^{\text{tip}}$, index MCP $\mathbf{k}_{\text{index}}^{\text{mcp}}$, thumb tip $\mathbf{k}_{\text{thumb}}^{\text{tip}}$, thumb MCP $\mathbf{k}_{\text{thumb}}^{\text{mcp}}$

**Output:** Gripper pose $(R_g, \mathbf{p}_g)$.

$\mathbf{i} \leftarrow \mathbf{k}_{\text{ind}}^{\text{tip}}, \ \mathbf{m} \leftarrow \mathbf{k}_{\text{ind}}^{\text{mcp}}, \ \mathbf{t} \leftarrow \mathbf{k}_{\text{thumb}}^{\text{tip}}, \ \mathbf{r} \leftarrow \mathbf{k}_{\text{thumb}}^{\text{mcp}}$   *// Extract keypoints*
$\mathbf{o} \leftarrow (\mathbf{t} + \mathbf{i})/2$   *// palm origin*
$v_z \leftarrow (\mathbf{i} - \mathbf{m})$   *// Z-axis direction*
$\bar{\mathbf{z}} \leftarrow v_z/(\text{NORMALIZE}(v_z) + 10^{-8})$
  *// Normalized Z-axis*
$v_y \leftarrow \text{CROSS\_PRODUCT}(\mathbf{i} - \mathbf{m}, \ \mathbf{m} - \mathbf{r})$
  *// Y-axis (palm normal)*
$\bar{\mathbf{y}} \leftarrow v_y/(\text{NORMALIZE}(v_y) + 10^{-8})$
  *// Normalized Y-axis*
$\bar{\mathbf{x}} \leftarrow \text{CROSS\_PRODUCT}(\bar{\mathbf{y}}, \bar{\mathbf{z}})$
  *// X-axis direction*
$\bar{\mathbf{z}} \leftarrow \text{SIGN}(\diamond) \cdot \bar{\mathbf{z}}$
$R_g \leftarrow \text{CONCATENATE}([\bar{\mathbf{x}}, \bar{\mathbf{y}}, \bar{\mathbf{z}}])$
  *// Rotation matrix*
$\mathbf{p}_g \leftarrow \mathbf{o}$   *// Position*
**Return** $(R_g, \mathbf{p}_g)$

## F  CONVERT HAND GESTURE TO ROBOT ARM

For the two different gesture types, namely whole-hand and finger-only, we designed corresponding mapping schemes to translate hand motions to a two-fingered gripper. For whole-hand gestures, our method primarily leverages keypoints on the palm plane to define the gripper's orientation and spatial position. For finger-only gestures, we incorporate fingertip positions to accommodate dexterous manipulation.

## G MORE VISUALIZATION RESULTS IN REAL-WORLD VALIDATION

Here, we present the experimental results for all designed tasks conducted on physical robot(UR5).

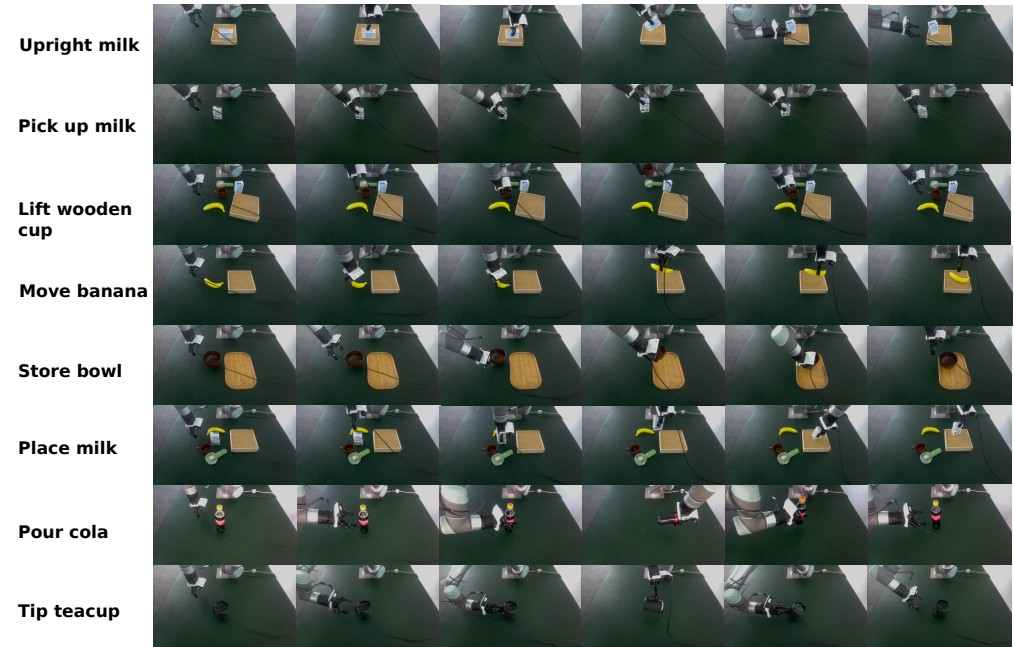

Figure 16: Visualization of 8 tasks on real robot

## H VISUALIZATION RESULTS OF MANIPTRANS

We mapped the hand gestures onto the dexterous hand and completed the training in a simulation environment. The training outcomes are shown in Appendix H.

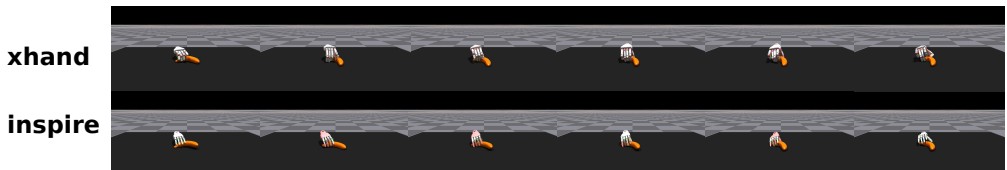

Figure 17: Visualization results of maniptrans

## I MOCAP HARDWARE SETUP

**Glove** The glove equipped with 16 Gen3 tactile sensors and 29 magnetic encoders. This glove is worn on a single hand and serves to collect high-frequency tactile data. The tactile sensors are capable of detecting pressure, force, and vibration, while the magnetic encoders are used to capture precise joint angles and movements of the fingers. This combination allows for detailed hand-object interaction data to be gathered with high temporal resolution.

**RGBD Cameras** Two Intel RealSense D455 RGBD cameras are used to capture depth and RGB data simultaneously. The cameras are mounted at strategic locations to ensure optimal coverage and accurate 3D spatial data. The depth cameras provide high-resolution depth maps, while the RGB cameras offer high-quality color images. For synchronized data acquisition, a synchronization cable is employed, connecting three cameras to ensure precise temporal alignment across all devices.



Figure 18: Mocap Handware Config Visualization

**RGB Cameras** A total of six high-resolution RGB cameras are used for detailed visual tracking. These cameras are positioned to cover different angles, enabling comprehensive capture of the environment and subjects. The cameras are used to provide complementary visual data to the depth information provided by the RGBD cameras.

## J EXPERIMENT DETAILS

We used the *RoboWheel*data to train four VLA/IL policies, namely ACT, DP, RDT-1B, and Pi0, to validate the effectiveness of our data.Before using these models for training, we first preprocessed the data to fit the observations, actions, and instructions $if\,needed$ required for model training.

### J.1 IMPLEMENTATION AND HYPER-PARAMETERS OF 4 VLA/IL POLICIES

For ACT ,we trained each task for 20,000 iterations, with 90% of the data used for training and the remaining 10% for validation. When training DP, we kept the same training steps, learning rate, and chunk size as ACT. The specific parameter values are listed in table 6.

| Hyperparameter (ACT) | Value | Hyperparameter (DP) | Value |
|---|---|---|---|
| Chunk_Size | 16 | Chunk_Size | 16 |
| Hidden_Dim | 512 | Action_Horizon | 8 |
| Batch_Size | 16 | Batch_Size | 16 |
| Learning_Rate | 1e-5 | Learning_Rate | 1e-5 |
| Dim_Feedforward | 3200 | Observation_Horizon | 8 |
| Num_Steps | 20000 | Num_Steps | 20000 |

Table 6: Hyperparameters of training ACT and DP

RDT was pretrained for 100,000 steps with a batch size of 8 per GPU on 4 GPUs, and all single-task fine-tuning was conducted for 10,000 steps with a batch size of 8 per GPU on a single GPUs.

Pi0 was pretrained for 100,000 steps with a batch size of 32 on 8 GPUs, and all fine-tuning was performed for 30,000 steps using the same batch size on a single GPU.

The other hyperparameters for RDT and Pi0 were kept consistent with the official documentation.

## K ARM ACTION REPLAY COMPARISON WITH DIFFERENT METHODS

Replay Comparison by Different Methods: We compared the mapping method of *RoboWheel* with YOTO and GAT-Grasp. YOTO and GAT-Grasp result in discrepancies in gripper position or orientation mapping, leading to failure, while *RoboWheel* provides more accurate and reasonable mapping.

## L LLM USAGE

We utilize large language models (LLMs) to polish our articles and correct errors, including grammatical mistakes and imprecise phrasing.

**GAT-Grasp**

**YOTO**

**Ours**

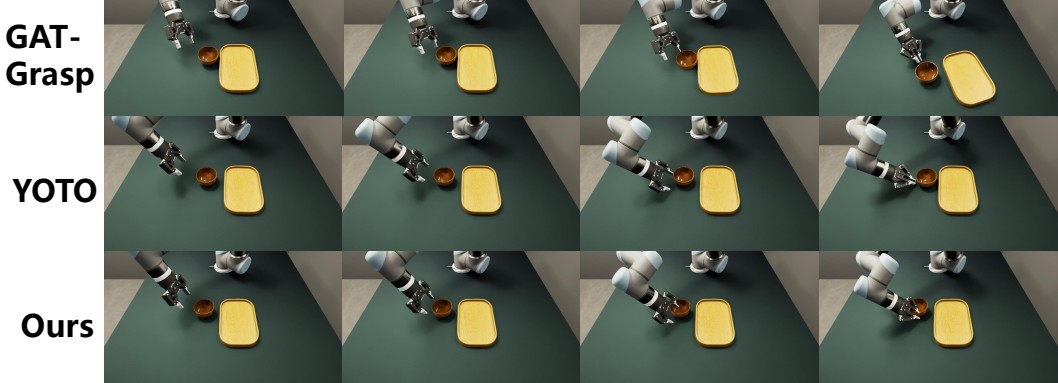

Figure 19: Replay Comparison by Different Methods: We compared the mapping method of *RoboWheel* with YOTO and GAT-Grasp. YOTO and GAT-Grasp result in discrepancies in gripper position or orientation mapping, leading to failure, while *RoboWheel* provides more accurate and reasonable mapping.

