# OpenReview forum: "ROBOWHEEL: A HELICAL DATA ENGINE FROM REAL-WORLD HUMAN DEMONSTRATIONS FOR CROSS-DOMAIN ROBOTIC LEARNING"
_ICLR.cc/2026/Conference — ICLR 2026 Conference Withdrawn Submission_

### Official Review · Reviewer_mKGk · 2025-10-26

**Soundness:** 3
**Presentation:** 3
**Contribution:** 3
**Rating:** 6
**Confidence:** 5

**Summary:**

This paper turns in‑the‑wild hand–object interaction (HOI) videos into physically consistent, executable robot trajectories and a large multimodal dataset that improves imitation and vision‑language‑action (VLA) policies across robot embodiments (arms, dexterous hands, humanoids). From RGB/RGB‑D, estimate hands/bodies (SMPL‑H/MANO) and 6‑DoF object pose/mesh; lift to a canonical world/action frame. Multi‑stage refinement with 2D reprojection losses, SDF‑based penetration/contact constraints, temporal smoothing, and a residual RL step to enforce reachability and plausible hand–object poses. Map refined hand motions to parallel‑jaw grippers, dexterous hands, and humanoids, producing both operational‑space (SE(3)) and joint‑space rollouts; robust gripper orientation via full‑hand vs. fingertip grasp categorization.

Simulation‑augmented “data flywheel”: In Isaac Sim, run HOI‑conditioned domain randomization (embodiment variants, object retrieval/replacement, background changes, left/right mirroring) and trajectory augmentations while preserving contact semantics—closing the loop for iterative policy improvement.

Key contributions:

Precision, physically plausible HOI reconstruction from monocular video and unified retargeting across embodiments.

A simulation‑augmented data engine that scales coverage and robustness while maintaining contact geometry.

A large‑scale multimodal dataset (~150k sequences) with RGB‑D, poses, contacts, partial tactile, language, and aligned robot actions.


Conclusion:
I think this is a solid system-level paper overall. However, several modules are ambiguous based on my system-level design experience. I’ve listed my questions below; if the authors address them, I will adjust my score accordingly.

**Strengths:**

Physically grounded reconstruction. Multi‑stage optimization: (I) 2D reprojection consistency, (II) SDF‑based contact/penetration handling with temporal smoothing, and (III) residual RL refinement for reachability and physical plausibility. This directly targets the classic gap between monocular estimation and robot‑usable contact trajectories.

Canonical action space + monocular SLAM alignment. Estimating intrinsics/extrinsics and mapping everything into a world frame / canonical action space should reduce viewpoint idiosyncrasies and stabilize retargeting across heterogeneous sources.

IK feasibility is handled via cuRobo with temporal seeding, which is a practical detail many pipelines skip.

Well‑thought simulation augmentation tied to HOI semantics. Object retrieval and replacement (Chamfer + AABB IoU + semantic embedding), background variation, embodiment variants, and trajectory adjustments (segment‑wise, contact‑aware transforms) keep contact frames consistent while diversifying observations.

**Weaknesses:**

Ablation depth is insufficient. The pipeline has many moving parts (2D losses, SDF penalties, temporal smoothness, residual RL refinement, canonical action space, object‑retrieval substitution, multiple augmentation knobs). It’s hard to isolate which stages contribute how much to the final gains. In particular, the RL refinement’s marginal value and cost aren’t disentangled.

Scale estimation & robustness for monocular‑only videos. When there’s no depth, metric scale is inferred via monocular depth prediction + back‑projection. Error sensitivity, per‑dataset breakdowns (RGB vs RGB‑D), and failure cases (thin/transparent/reflective objects; fast motion; severe occlusion) aren’t quantified.

Dexterous & humanoid validation is thin. The paper describes retargeting to dexterous hands and humanoids, but the strongest quantitative results and replays emphasize two‑finger grippers. Evidence of real‑robot performance on dexterous/humanoid platforms (beyond sim) is limited or absent.


Task/domain coverage. The current focus is rigid objects and single‑hand/single‑arm manipulation. Deformables, tools with complex dynamics (e.g., cutting), liquids beyond pouring, and bimanual coordination are not demonstrated

**Questions:**

When only monocular RGB (no depth) is available, how accurate is metric scale recovery? Please report scale error distributions and reconstruction metrics split by RGB‑D vs RGB‑only inputs.

How sensitive is the joint optimization to SDF quality? Do you fail on thin/concave/reflective/translucent objects? Any fallback for poor meshes?

What are typical failure modes (e.g., severe occlusion, fast hand motion, motion blur), and how frequently do they occur?

Is the RL policy trained per‑clip (online fitting) or amortized across clips? What are the sample budgets, training time, and compute for stage (B)?

---

### Official Review · Reviewer_QcHj · 2025-10-30

**Soundness:** 2
**Presentation:** 1
**Contribution:** 3
**Rating:** 4
**Confidence:** 3

**Summary:**

The paper presents a data engine that converts Internet and real-world HOI videos into robot-usable supervision transferable across embodiments such as robotic arms, dexterous hands, and humanoids. The multi-stage pipeline start from video to 1) HOI reconstruction; 2) physical refinement 3) cross-embodiment retargeting 4) simulation augmentation 5) training, which is iteratively applied to enhance policy performance. The resulting dataset includes over 150K trajectories after augmentation, with multimodal data. Experiments show improved reconstruction metrics over prior work, comparable policy performance to teleoperation data, enhanced generalization through augmentation.

**Strengths:**

1. Clear decomposition of the overall pipeline and the decoupling is reasonable.
2. Introduction of a large-scale, multimodal dataset (RGB-D, object assets/poses, contact state, tactile)
3. Physically plausible formulation combining contact/penetration optimization with RL residual refinement, an emerging and interesting direction.
4. Augmentation improves data coverage across multiple arms for retargeting; and the paper shows potential extension to dexterous hands and humanoids, though not fully verified.

**Weaknesses:**

The paper presents a highly complicated paradigm with multiple interconnected modules, yet each module (data acquisition, reconstruction, retargeting, and RL refinement) is insufficiently explained and lacks clear verification or ablation. The overall contribution becomes hard to isolate due to the lack of module-wise validation.

* No data demonstration video is provided in the supplementary material. Video demos are critical for verifying HOI reconstruction quality, assessing motion naturalness and contact realism, and evaluating the retargeting performance, and the validity of the proposed augmentation strategy. And those demo should be presented with examples at scale to verify the effectiveness when scaling up to whole dataset

* Although the paper reports over 150k trajectories covering diverse embodiments and modalities, the dataset details remain unclear: release status, licensing terms, and provenance of each subset are not specified. Without public availability or documentation, reproducibility, transparency, and community adoption are severely limited.

* The pipeline relies on several heavy and failure-prone components, but the paper does not analyze robustness to component failures, accumulated error propagation, or the engineering cost required to operate the full system at scale.

* Specifically for the residual RL refinement part, (many other parts are underexplained as well): only two simple grasping examples are shown, and no quantitative metrics such as success rate or convergence behavior are reported. It remains unclear how large-scale RL imitation training is executed given the large dataset size, how failed rollouts are handled, or whether the policy generalizes beyond toy cases. The physical simulation setup lacks transparency. Critical parameters, such as contact stiffness, friction coefficients, and object mass are not provided, leaving uncertainty about how closely the simulated dynamics reflect real-world physics.

* No verification of humanoid retargeting

**Questions:**

Inspire hand being an underactuated hand, requires explicit modeling of joint coupling and limited controllability. The paper does not clarify how underactuation is represented in the simulator or whether the model captures the physical constraints that arise in real hardware.

**Details Of Ethics Concerns:**

Ethics Statement notes restriction to academic use, but the paper should clarify consent, licensing, and jurisdictional compliance for both the collected in-the-wild videos and the public HOI datasets

---

### Official Review · Reviewer_NCsS · 2025-11-01

**Soundness:** 3
**Presentation:** 2
**Contribution:** 2
**Rating:** 2
**Confidence:** 5

**Summary:**

This work describes RoboWheel, a "helical" data generation pipeline for converting human hand-object demonstration videos into robot hand and object pairs.

The overall pipeline has 3 stages:
1. **Stage 1:** The authors use off-the-shelf models to extract human hand poses, body poses, camera parameters, and object geometry.
2. **Stage 2:** The authors refine the Stage 1 outputs. They run a contact-aware optimization, then use the resulting outputs to define a reward which they train an RL policy on.
3. **Stage 3:** The authors retarget the refined hand-object parameters to various robot embodiments: parallel jaw, dexterous hands, and humanoid robots. They also apply workspace and object mesh augmentations.

The authors present a dataset, RoboWheel-150k, which is the output of this pipeline on a combination of internet videos, public HOI datasets, and self-collected mocap data. This data appears much larger than previous datasets, and is unique in that it includes both tactile/force data and contact state.

Finally, the authors validate their dataset through a combination of reconstruction metrics and behavior cloning results.

**Overall rating:** given the weaknesses below, I'm currently rating the paper as a *reject*. However, I'm more than happy to revisit this based on responses or updates from the authors.

**Strengths:**

Overall, the paper seems well-written and timely. Gathering more precise, in-embodiment, and physically plausible hand-object interaction sequences seems very valuable for the robot learning community. The pipeline of running off-the-shelf vision models, optimizing them for consistency, and ensuring physical plausibility with RL seems very reasonable. The robot manipulation experiments with a parallel jaw gripper verify that the generated data can actually be used to train useful policies.

**Weaknesses:**

The paper seems to be missing many details, which are especially important for a pipeline/dataset contribution. See "Questions".

The paper includes retargeting for dexterous hands and humanoid robots. However, there aren't experiments to validate that this data is high-enough quality to be used for robot policy learning.

Finally, I found the paper somewhat lacking in actually showing examples of the data that's generated by the system. It would be nice to see correspondence between input human demonstrations and outputted robot demonstrations.

**Questions:**

I'm not really convinced about the "helical" or "wheel" description of this work. The pipeline appears to be one-way retargeting: the input is human video, the output is retargeted robot trajectories. Can you explain where the loop come from? In Figure 1, there's an arrow pointing from the robot dataset up toward HOI data sources, it's unclear to me how this arrow manifests in the work.

On L204-205, the authors claim to use DROID-SLAM to estimate camera intrinsics and extrinsics. However, DROID-SLAM doesn't estimate intrinsics, it assumes known intrinsics. How are intrinsics being estimated? This seems very important for physically plausible reconstructions.

The paper mentions tactile/force and contact state, but it's not explained how these signals are produced. Where do they come from? Are they available in all sequences (extracted also from the internet data) or only the mocap data?

The RoboWheel pipeline has many hand-designed stages. It would be nice to know failure rates for each stage. How often does the pipeline work for in-the-wild data? Does the data need to be filtered? Are there cases where the retargeting from human hand to robot hand (parallel jaw gripper or dexterous) fails due to the embodiment gap?

The paper states that RoboWheel-150K combines internet videos, public HOI datasets, and self-collected mocap data. Can the authors provide a quantitative breakdown of how many sequences come from each source? This would help assess data quality, diversity, and reproducibility.

There's a VLM that does success prediction in Figure 2. Is this described in the text? I couldn't find it.

Do you have a breakdown of how much data of each type is in the dataset? Counts by source, modality, robot types, tasks, etc would be appreciated.

---

### Official Review · Reviewer_nRSk · 2025-11-03

**Soundness:** 2
**Presentation:** 2
**Contribution:** 3
**Rating:** 4
**Confidence:** 3

**Summary:**

The paper presents RoboWheel, an approach that converts videos of human-object interaction into data ready to be used for cross-morphology robotic learning. The paper describes the complete system, from reconstructing the hand-object interaction videos, retargeting these trajectories to different robot embodiments and the strategies for augmentation in simulation when training the policies. The approach is demonstrated with real-world experiments.

**Strengths:**

- The paper presents a complete pipeline for data collection and policy training. I appreciate the end to end nature of the effort.
- The proposed reconstruction approach is demonstrated on HΟ-Cap with promising results compared to previous baselines.
- The results in Table 3 show the value of using the large scale RoboWheel data.

**Weaknesses:**

- The paper makes claims about processing Internet/in-the-wild and converting them to trajectories to be included in the dataset. As far as I can tell, there is no information about this data (how many videos, what is their length, etc).
- There are no video results demonstrating the success of the method on in-the-wild videos.
- One of the key contributions of the paper is proposing the RoboWheel dataset. Unfortunately, there is no detailed breakdown about where the RoboWheel data come from. There is a statement that 150k trajectories (elsewhere 150k frames) are available, but there are no further details.
- The paper promises a flywheel, but I don't think it really demonstrate the continuous nature of the data collection. In Figure 1, I see an arrow pointing to HOI sources, but I am not sure how this is executed.
- Minor: In Ln 879, the correct reference is missing.

**Questions:**

- I am very interested in more details about the FlyWheel data. What is the breakdown? What are the actual sources? In particular, about the in-the-wild videos, the paper needs more details, related to the instances, success rate, examples of successful reconstructions, etc. The intro makes specific claims about using Internet/in-the-wild videos, but the paper does not follow up on that.
- Could you clarify the "task-specific finetuning" of the +5kRW policies (Ln 421)? What does that exactly mean?

---

### Note · Authors · 2025-11-14

**Comment:**

We request withdrawal of this submission. All co-authors have reviewed and consent to this withdrawal.

**Withdrawal Confirmation:**

I have read and agree with the venue's withdrawal policy on behalf of myself and my co-authors.